# Psychological Profiles in Ulcerative Colitis and Crohn’s Disease: Distinct Emotional and Behavioral Patterns

**DOI:** 10.3390/biomedicines13071694

**Published:** 2025-07-10

**Authors:** Antonio Maria D’Onofrio, Eleonora Maggio, Valentina Milo, Gaspare Filippo Ferrajoli, Daniele Ferrarese, Daniela Pia Rosaria Chieffo, Massimiliano Luciani, Antonio Gasbarrini, Gabriele Sani, Franco Scaldaferri, Rosaria Calia, Giovanni Camardese

**Affiliations:** 1Section of Psychiatry, Department of Neuroscience, Università Cattolica del Sacro Cuore, 00168 Rome, Italy; antoniomdonofrio@gmail.com (A.M.D.);; 2Unit of Clinical Psychology, Fondazione Policlinico Universitario Agostino Gemelli IRCCS, Università Cattolica del Sacro Cuore, 00168 Rome, Italy; 3Dipartimento di Salute Mentale, Ospedale di Latina, ASL Latina, 04100 Latina, Italy; 4CEMAD (Digestive Disease Center), Department of Medical and Surgical Sciences, Policlinico Universitario “A. Gemelli” IRCCS Foundation, 00168 Rome, Italy; 5Gastroenterology and Liver Unit, Department of Internal Medicine, School of Medicine, Catholic University of the Sacred Heart, A. Gemelli Foundation, 00168 Rome, Italy; 6Section of Psychiatry, Department of Neuroscience, Fondazione Policlinico Universitario Agostino Gemelli IRCCS, 00168 Rome, Italy; 7Department of Life Science, Health, and Health Professions, Link Campus University, 00165 Rome, Italy

**Keywords:** ulcerative colitis, Crohn’s Disease, inflammatory bowel disease, emotional patterns, behavioral pattern

## Abstract

**Background/Objectives**: Ulcerative colitis (UC) and Crohn’s disease (CD) are two forms of inflammatory bowel disease (IBD), which, despite their shared inflammatory nature, differ markedly in clinical presentation and disease course. In this study, we aimed to explore whether these clinical differences are also reflected at the psychological level. Specifically, we sought to delineate the personality characteristics of a sample of patients with IBD and to investigate psychological and psychopathological differences between individuals with UC and CD. **Methods**: We enrolled 29 (44.61%) UC patients and 36 (55.39%) CD patients, all aged 18 years or older. Each participant completed the Minnesota Multiphasic Personality Inventory-2 (MMPI-2), which was subsequently scored and interpreted by trained psychologists. The MMPI-2 is a 567-item inventory with dichotomous answers (true/false), providing measures of a wide range of symptoms, beliefs, attitudes, and personality traits. **Results**: The total sample showed clinically significant elevations on hypochondriasis (Hs), health concerns (HEA), general health concerns (HEA_3_), and physical malfunctioning (D_3_) scales. UC patients had statistically significant higher scores on hypomania (*p* = 0.043), lack of ego mastery—defective inhibition (*p* = 0.006), and fears (*p* = 0.038) scales than CD patients. On the other hand, CD patients showed statistically significant higher scores on the Overcontrolled Hostility scale (*p* = 0.043). **Conclusions**: Both groups of patients experience emotional difficulties related to their clinical conditions, leading to an increased preoccupation with bodily symptoms and illness. These aspects appear to be accompanied by shifts in mood towards a more depressive state. Notably, the UC group demonstrates a greater degree of impairment compared to the CD group, with experiences of anxiety, stress, difficulties in emotional control, and emerging relational challenges.

## 1. Introduction

Inflammatory bowel disease (IBD) includes two chronic inflammatory disorders: Crohn’s disease (CD) and ulcerative colitis (UC). The well-being and daily life of individuals with IBD can be notably influenced by extraintestinal manifestations commonly target joints, skin, eyes, liver, lungs, and pancreas [1]. IBD patients experience a lower Quality of Life (QoL) compared to healthy individuals, mainly during active disease and in those with CD [2]. According to a systematic review by Neuendorf et al., the pooled prevalence of anxiety disorders among patients with IBD was 20.5%, while the prevalence of anxiety symptoms was higher, at 35.1%. Patients with active disease showed a markedly greater prevalence of anxiety symptoms (75.6%) compared to those in remission. The pooled prevalence of depressive disorders was 15.2%, and 21.6% for depressive symptoms. Depressive symptoms were more frequent in patients with CD (25.3%) compared to those with UC, and were also more common in individuals with active disease (40.7%) than in those in remission [3]. Persistent personality characteristics in IBD patients include heightened conscientiousness, perfectionism, increased sensitivity, and a sense of inadequacy in performance. Moreover, females with IBD are further marked by gender role conflict and lower assertiveness [4].

There are only very few studies, and they are now quite dated, that have employed a rigorous psychodiagnostic evaluation like Minnesota Multiphasic Personality Inventory-2 (MMPI-2) to analyze the personality characteristics of patients with IBD. In a study conducted by West (1970) [5], UC patients were distinguished from individuals with various conditions deemed psychosomatic at the time. UC patients exhibited significantly lower psychological disturbance compared to the broader category of “psychosomatic patients.” Their MMPI profile demonstrated a moderately elevated neurotic configuration, resembling that of medical patients in general. In a study by McMahon et al. (1973) [6], IBD patients had higher mean score on two of the three scales within the “neurotic triad” (Hs = hypochondriasis; D = depression; Hy = hysteria) than their healthy siblings. This difference reflects traits associated with neuroticism, such as a tendency toward anxiety, persistent worries, mood fluctuations, and a prevalence of negative emotions. In a study conducted by Liedtke et al. (1977) [7] UC patients had more elevated scores on scales related to hypochondriasis, depression, paranoia, and social introversion than healthy controls. In a study by Gathmann et al. (1981) [8], patients diagnosed with UC patients exhibited heightened levels of anxiety and neuroticism, along with a tendency toward introversion.

Considering the guidelines [9,10] and potential psychological implications associated with these conditions, it is necessary to systematically investigate the psychological symptoms and personality structure of patients with IBD. While both CD and UC share overlapping features, they differ substantially in their clinical presentation and disease management [11]. CD can affect any part of the gastrointestinal tract, often leading to complications such as fistulas and strictures [12], whereas UC is limited to the colon and rectum, with inflammation typically confined to the mucosal layer [13]. These differences have profound implications for patients’ quality of life [14]. For example, individuals with CD may face greater challenges due to unpredictable disease patterns and the need for surgical interventions [15], whereas those with UC may experience a more continuous disease course [16], impacting medication adherence and coping strategies differently. Moreover, the presence of depressive symptoms in IBD patients has a direct impact on gut inflammation, and it is a strong risk factor for subsequent clinical deterioration [17].

The principal aim of this study is to explore psychological and psychopathological differences between patients with Crohn’s disease and ulcerative colitis. By comparing these groups, we aim to uncover how the distinct clinical characteristics may be reflected in their psychological profiles and personality traits, ultimately providing new insights for targeted therapies aimed at achieving the overall well-being of the patient. To the best of our knowledge, this is the first study directly comparing personality traits between individuals with CD and UC.

## 2. Materials and Methods

### 2.1. Patients and Study Design

We recruited consecutive patients affected by UC and CD who were at least 18 years old and attending to the IBD Unit of CEMAD (Center for Digestive Disease) of “A. Gemelli” IRCCS Hospital in Rome. Only patients referred by the gastroenterologist to a psychologist within the Psychogastroenterology Service were included in the study; these referrals occurred either upon the patient’s request for brief psychotherapy sessions or when deemed appropriate by the gastroenterologist. The patients were enrolled between 2018 and 2019. Inclusion criteria were age > 18, diagnosis of UC/CD, at least a level of education of lower secondary school diploma, acceptance of the informed consent, and willingness to complete the MMPI-2 assessment. We excluded patients with age > 85 and with a diagnosis of psychosis or dementia. The enrolled patients with a histologically confirmed diagnosis of UC or CD underwent a clinical interview and a physical examination with the gastroenterologist, which included the collection of personal data, routine demographics, extent and duration of disease, disease activity assessments, previous biologic and immunosuppressive drugs, and concomitant medication. All participants filled the MMPI-2 that was then scored and interpreted by trained psychologists. All procedures complied with the informed consent administered to the patients that accepted a complete assessment of their personality after the gastroenterologist’s visit. On average, the duration of the administration of MMPI-2 was about 60 min. Before proceeding with the filling of the questionnaire, each patient received indications about the methods of compilation, it was specified that there were no right or wrong answers and that they had to respond in the sincerest way possible. The patients were encouraged to fill all the items without omissions.

The identity of the participants was maintained confidential by associating each patient with an alphanumeric code. In this ecological study, no formal sample size calculation was performed. The study population was defined based on available data from patients meeting the inclusion criteria, reflecting real-world conditions rather than a predetermined sample size.

Ethical approval for this study was obtained from local ethical committee (ID1886/Prot. N. 0011626/18).

### 2.2. Methodology

All patients underwent the MMPI-2, the most widely researched and used clinical instrument in the field of personality assessment. We did not perform an independent validation of the MMPI-2 within our sample and the scales were administered and scored in accordance with the procedures and normative data described by Butcher et al. (2001) [18]. MMPI-2 was validated empirically recruiting patients with clear psychopathological diagnoses compared with samples of non-patients. It is currently available and standardized in Italian [19]. MMPI-2 is an inventory composed of 567-item with dichotomous answers (true or false) that provides measures of a broad variety of symptoms, beliefs, attitudes, and personality characteristics. It is standardized with uniform T-scores; in particular, clinical (or “high”) scores are T > 65. It consists of 9 validity scales that evaluate people’s attitude to the test, self-presentation, degree of cooperating with the testing and accuracy of answers. The main scales of MMPI-2 are the 10 clinical scales related with high risk of having a mental health disorder: hypochondriasis (Hs), depression (D), hysteria (Hy), psychopathic deviation (Pd), paranoia (Pa), psychasthenia (Pt), schizophrenia (Sc), hypomania (Ma), masculinity/femininity (Mf), and social introversion (Si). Moreover, there are 31 Harris and Lingoes subscales for clinical scales, useful for interpreting clinical scales and with more homogeneous content. In addition, there are 15 content scales and 27 content component scales to clarify the meanings of clinical scales. In total, 16 supplementary scales about specific areas provide information not available from clinical scales and are divided into four groups: generalized emotional distress scales, broad personality characteristics scales, behavioral dyscontrol, and gender role. Lastly, the Personality Psychopathology Five (PSY-5) scales measure personality traits related with more long-lasting psychopathology (aggressiveness, psychoticism, discontraint, negative emotionality/neuroticism, and introversion/low positive emotionality). More information about the meaning of all the scales of MMPI-2 is reported in the Appendix A.

### 2.3. Statistical Analysis

All statistical analyses were performed using SPSS software (Version 26.0. IBM Corp: Armonk, NY, USA). The sample was described in its whole characteristics by descriptive statistical techniques. Quantitative data were described by mean and standard deviation (SD), whereas qualitative as absolute and relative percentage frequency. To verify Gaussian distribution of quantitative variables, the Shapiro–Wilk test was applied. The comparisons of discrete variables between groups were deployed based on the Chi-squared index, while ANOVA test was performed for continuous variables. *t*-test was performed to evaluate the differences in the MMPI-2 scales’ mean score between UC and CD patients. Multivariate analysis of covariance (MANCOVA, covariates: age and gender) and Bonferroni post hoc comparison were performed on clinical scales, Harris and Lingoes subscales, and content scales to evaluate if age and gender could influence our results. We calculated the percentage of patients who scored above the clinical cut-off only in scales that showed statistically significant differences between the two groups.

### 2.4. Data and Code Availability Statement

The data, code, and materials supporting the findings of this study are available upon reasonable request from the corresponding author. Due to privacy and ethical considerations, some data may be restricted.

## 3. Results

### 3.1. Descriptive Sociodemographic Data and Clinical Data

In the following table, we summarize UC and CD patients’ sociodemographic and clinical data. We have defined disease activity using the Mayo Clinical Score [20] for patients with UC and the Harvey–Bradshaw Index for patients with CD [21]. For biologic therapy, we mean the administration of drugs such as infliximab, adalimumab, etc., in addition to anti-inflammatory or corticosteroid medications (Table 1).

### 3.2. Personality Characteristics of UC and CD Patients

In the Appendix A, we summarize UC and CD patients’ psychometric data about all the scales and the subscales of MMPI-2 (Appendix A). A score of T > 65 is considered clinically significant; a score of T > 60 is moderately high. Validity scales highlight that both UC and CD groups present adequate scores; they responded accurately and consistently to items with only a few omissions; they did not make any particular attempts consciously or unconsciously to alternate results to provide an unrealistic self-image. Therefore, considering the median T scores in validity scales, there are no signs of simulation or dissimulation, resistance, or other forms of censorship.

UC patients show clinically significant elevations on the hypochondriasis (Hs = 66.66 ± 11.23), health concerns (HEA = 66.86 ± 10.03), general health concerns (HEA_3_ = 67.62 ± 11.11), and physical malfunctioning (D_3_ = 65.05 ± 13.68) scales. Moreover, they have moderately high scores on depression (D = 63.21 ± 14.88), subjective depression (D_1_ = 61.37 ± 14.48), hysteria (Hy = 61.17 ± 14.47), lassitude malaise (Hy_3_ = 64.17 ± 12.63), somatic complaints (Hy_4_= 62.66 ± 12.99), psychasthenia (Pt = 61.38 ± 12.25), schizophrenia (Sc = 60.69 ± 10.69), lack of ego mastery—cognitive (Sc_3_ = 60.64 ± 14.68), bizarre sensory experiences (Sc_6_ = 63.53 ± 12.18), persecutory ideas (Pa_1_ = 60.47 ±12.23), anxiety (ANX = 62.97 ± 12.40), neurological symptoms (HEA_2_ = 64.21 ± 8.99), college maladjustment (Mt = 61.34 ± 11.94), and post-traumatic stress disorder—Keane (PK = 61.97 ± 13.35).

On the other hand, CD patients show clinically significant elevations on the hypochondriasis (Hs = 67.67 ± 11.86), health concerns (HEA = 66.75 ± 10.67), general health concerns (HEA_3_ = 68.00 ± 11.79), and physical malfunctioning (D_3_ = 66.00 ± 12.84) scales. Also, they have moderately high scores on the depression (D = 63.69 ± 14.60), subjective depression (D_1_ = 60.66 ± 15.07), hysteria (Hy = 63.47 ± 12.30), lassitude malaise (Hy_3_ = 62.73 ± 13.35), somatic complaints (Hy_4_ = 64.00 ± 12.16), and neurological symptoms (HEA_2_ = 62.42 ± 11.63) scales.

### 3.3. Differences Between UC and CD Patients in the MMPI-2 Scales Scores

The groups differed in gender (χ^2^ = 4.19; *p* = 0.040) but not in age (F_(2)_ = 223.922; η^2^ = 0.50; *p* = 0.824).

After performing a *t*-test for all the scales of MMPI-2, we found that UC patients had statistically significant higher scores on the hypomania (*p* = 0.043), lack of ego mastery—defective inhibition (*p* = 0.006), and fears (*p* = 0.038) scales than CD patients. On the other hand, CD patients showed statistically significant higher scores on the overcontrolled hostility scale (*p* = 0.043) (Figure 1 and Table 2).

Considering only statistically significant differences in the scales between the two groups, in Figure 1, we reported the percentages of patients who scored over the clinical cut-off (T > 65). In particular, 31% of UC patients (vs. CD = 2.80%) had clinical scores on hypomania, 27.60% (vs. CD = 8.30%) on lack of ego mastery—defective inhibition, and 17.20% (vs. CD = 11.10%) on fears. In total, 5.60% of CD patients showed clinical scores on the overcontrolled hostility scale (vs. UC = 0%).

After conducting MANCOVA analysis, we highlighted that no statistical significance of the covariate age was highlighted on the MMPI-2 scales. Gender was found to be statistically significant covariate in terms of the group comparison of the lack of ego mastery—defective inhibition (Sc5) (F_(1)_ = 7.92; *p* = 0.007) and fears (FRS) (F_(1)_= 6.09; *p* = 0.016) scales, in which UC patients had statistically significant higher scores (M_Sc5_ = 60.00; M_FRS_ = 58.97) than CD patients (M_Sc5_ = 51.64; M_FRS_ = 53.33).

## 4. Discussion

The principal aim of this study was to explore with MMPI-2 the personality traits and psychopathology of patients with chronic idiopathic inflammatory diseases like CD and UC. We used the MMPI-2’s interpretive norms to discuss the observed profile patterns; however, due to the study’s cross-sectional design, these interpretations indicate only associations and do not support causal inferences.

Both CD and UC are characterized by a persistent disease course marked by intervals of remission and episodes of active intestinal inflammation. Given their chronic nature, unpredictable trajectory, onset typically at a younger age, and the requirement for costly medical and surgical interventions, IBD poses a significant public health challenge impacting patients’ health-related quality of life [22].

Both the UC and CD groups have demonstrated a consistent and comprehensive completion, providing a realistic self-view without aspects of underreporting or overreporting. For this reason, the personality profiles are valid. Both groups display difficulties in terms of symptoms and psychological functioning and show clinically significant elevations in scales highlighting problems and emphasize physical discomfort. In fact, they report increases in the hypochondriasis (Hs) clinical scale, which describes individuals who have broad and nonspecific somatic complaints and may develop somatic symptoms in reaction to stress. They also score higher on the health concerns (HEA) content scale, which is used to assess health status and the prevalence of medical conditions. This is demonstrated by the clinical subscales of physical malfunctioning (D_3_), somatic complaints (Hy_4_), lassitude malaise (Hy_3_), and general health concerns (HEA_3_) content, all of which are related with physical issues and an abnormal concern about health. It is also noteworthy that both groups exhibit a moderate elevation in the neurological symptoms (HEA_2_) content subscale, indicating impairments in sensory-motor experiences typically associated with neurological disorders. Our results are coherent with the previous literature, which has highlighted that neuroticism, that reflects feelings of distress, has a significant negative correlation with QoL in IBD patients and may predisposes them to perceive situations as threatening, leading to misinterpretations of somatic symptoms [23]. Worries about health are important core personality characteristics in IBD patients and health anxiety is probably related with fear of future disease. Individuals with UC and CD encounter a 2-to-2.5-fold increased risk of colorectal cancer compared to the general population and their risk of mortality associated with this malignancy is approximately 1.5 times higher than in the general population [24]. In fact, another psychological characteristic observed in IBD patients is alexithymia, marked by difficulties in identifying and expressing emotions, reduced empathy, and a tendency to experience emotions somatically [25]. Studies show elevated Toronto Alexithymia Scale scores in both IBS and IBD patients compared to controls [8], further emphasizing the interplay between psychological factors and physical health in these conditions.

Regarding mood dimension, both groups exhibit moderately elevated scores on the depression (D) clinical scale and in the subjective depression (D_1_) clinical subscale, which describes the tendency to experience negative emotions and fundamentally serves as an indicator of discomfort and a sense of dissatisfaction with one’s life conditions, suggesting a predisposition to mood fluctuations. This result is consistent with the literature, which showed that levels of depression and anxiety were higher in CD and UC patients than in healthy controls with large effect sizes. In a global survey, it was highlighted that rectal urgency was reported in 20.2% of patients with UC and in 16.4% with CD [26]; it is crucial to consider the psychological impact of this symptom in IBD patients, in particular as regards depression symptomatology, feeling judged or misunderstood because of the symptoms might lead patients to mood deflection and isolation, creating a vicious circle whereby disease activity leads to a set of thoughts and behaviors which trigger more disease activity and vice versa [27]. Depressive symptoms are important predictors of future clinical deterioration in IBD, significantly increasing the risk of flares and new extra-intestinal manifestations [17], as a consequence it is important to improve integrated treatment strategies for these diseases.

Both UC and CD patients showed moderately high scores on the clinical scale of Hysteria (Hy), indicating a tendency towards specific psychosomatic disorders characterized by pains and complaints. Simultaneously, this scale explores a lack of introspection, highlighting a stress response style that tends to express conflicts and/or responsibilities considered difficult to manage through the body. Somatic symptoms gradually tend to replace emotions and lead to a “psycho-somatization of affects” that impoverishes patient’s emotional life [28].

Our results confirm that IBD is associated with a range of psychological challenges; this is consistent with a recent study by Gabova et al. (2024) [29], in which emerged five critical topics in IBD patients: sexual activity, body image and discomfort, partner relationships, family planning, and the role of gastroenterologists in family planning decisions. These results underscore the impact of IBD on quality of life, emphasizing the necessity for proactive information dissemination and open communication from healthcare professionals.

After analyzing the core psychopathological characteristics of IBD, the second aim of this study was to identify differences between those groups and peculiar personality traits of them, to understand how different IBD could be related with psychological symptomatology.

UC patients reveal moderate elevations in the clinical scale of Psychasthenia (Pt), that describes individuals characterized by experiences of anxiety, fear, and excessive worries. Moderate scores on this scale denote the presence of a high level of anxiety and tension, restlessness, and emotional excitability. These aspects of anxiety and tension are also evident in the moderate elevation of the anxiety (ANX) content scale, which assesses the presence of negative effects such as nervousness, restlessness, dysphoric mood tones, and insecurity, along with the sensation of being overwhelmed by daily responsibilities. Furthermore, the increase in the post-traumatic stress disorder (PK) supplementary scale suggests the presence of maladjustment and a generalized emotional disturbance [30]. At the cognitive level, the elevation of the schizophrenia (Sc) scale, due to its heterogeneity, describes both misinterpretations of reality and behavioral manifestations of withdrawal and rigidity in emotional responses. Moderate elevations in scores tend to depict individuals exhibiting restricted affectivity and emotional detachment. UC patients show a moderate score in the lack of ego mastery-cognitive (Sc_3_) clinical subscale, reaffirming a perception of difficulties in basic neuropsychological functions such as attentive difficulties and peculiar thought processes. This aspect is further supported by the elevation of the bizarre sensory experiences (Sc_6_) subscale, which measures unusual somatic experiences and bizarre sensory perceptions. Finally, through the moderate elevation of the persecutory ideas (Pa1) subscale, there is a tendency to interpret the actions of others as humiliating or threatening. This is coupled with an inclination to attribute aspects of personal discomfort to external factors and a belief in receiving less understanding from the social and friendly context.

Considering CD patients’ group, there is no further information regarding their personality beyond what has been reported for the entire sample. Regarding the comparison between the two groups of IBD patients, it emerged that the UC group shows a higher score on the Mania (Ma) scale, highlighting greater physical and mental energy, presence of mood irritability compared to patients with CD. Another significant difference appears in the lack of ego mastery—defective inhibition (Sc5) subscale, indicating that UC patients have a greater sensation of not having control over their impulses. They experience their emotions as something strange and alien, occasionally feeling vulnerable to their own feelings. This result is in contrast with a previous study in which it was highlighted that CD patients presented higher impulsive sensation seeking than UC patients [31]. This apparent discrepancy can be understood by considering the different constructs being assessed. The Sc5 subscale of the MMPI-2 reflects a sense of psychological disorganization and diminished control over internal emotional states, rather than outward-directed impulsive behavior. In contrast, impulsive sensation seeking, described by Hyphantis et al. [31], is more aligned with externalizing tendencies and behavioral impulsiveness, often associated with novelty seeking and risk-taking behaviors. Overall, the two findings may not be mutually exclusive, but rather reflect different dimensions of emotional dysregulation and impulsivity within UC and CD populations. UC patients exhibit significantly elevated scores on the fears (FRS) scale, increased specific fears, phobias, and anxious responses tied to the unpredictability of external events. Conversely, CD patients scored higher exclusively on the overcontrolled hostility (O-H) scale, indicating a tendency to suppress anger and emotional distress. While driven by different underlying mechanisms, both psychological profiles contribute to delayed help-seeking: in UC, anticipatory anxiety may prompt symptom avoidance and reluctance to consult healthcare providers [32,33]; in CD, emotional repression often leads individuals to internalize suffering rather than voicing their discomfort. Such delays can prolong disease management and adversely impact clinical outcomes. Elevated anxiety and phobia in IBD are associated with increased symptom severity [34], reduced quality of life [35], long-term adverse disease outcomes [36], and postponed medical consultation [37]. At the same time, heightened anxiety and phobic reactivity may also lead to increased unscheduled or symptom-driven contact with healthcare services [38,39], reflecting emotional dysregulation, health-related worry, and misinterpretation of benign symptoms. In CD, impaired emotional processing—including avoidance and suppression—has been shown to mediate fatigue and exacerbate disease activity [40,41].

The findings highlighted so far demonstrate how both these groups of patients experience emotional difficulties that appear to be linked to clinical conditions, resulting in an increased focus on the body and illnesses. These aspects seem to be accompanied by mood changes in a depressive way. However, the UC group shows a greater impairment compared to the CD group, in which, in addition to the already mentioned elements, experiences of anxiety, stress, difficulties in emotional control, and relational aspects emerge. The differences between the two groups highlight how the UC group exhibits greater fears and difficulties in emotional control.

This study presents some limitations. Sample size of UC and CD patients is relatively small; in future research, it would be interesting to analyze personality on a larger cohort of patients with IBD. Moreover, personality was assessed at a single point in time—referring to stable traits observed over years rather than transient states—since the MMPI-2 is designed to capture enduring trait-level characteristics. Therefore, future studies would benefit from exploring how IBD may influence psychopathological features over longitudinal intervals, particularly following surgical interventions or major clinical changes that significantly impact daily life. Such investigations could help clarify the role of personality traits in the development and maintenance of IBD. The MMPI-2 is an older instrument; however, at the time of its administration in Italy, the MMPI-3 had not yet been published. We acknowledge the absence of biological data—such as inflammatory cytokines, stress-related hormones —as a limitation, since it precludes the possibility of exploring potential correlations between psychopathological features and relevant biological markers. Moreover, the study did not include healthy controls or patients with other chronic diseases as comparison groups, making it impossible to determine whether the observed psychological characteristics are specific to patients with IBD. As such, the section describing the psychological features of the entire sample should be interpreted as purely descriptive. Even though we performed a MANCOVA to control the effect of gender on the MMPI-2 scales, we acknowledge that future studies would benefit from more gender-balanced cohorts of UC/CD to fully validate these findings.

Consistent with our results, it is important to explore personality traits in patients with digestive tract diseases in order to understand how personality traits could be related to IBD symptoms and, as a consequence, to disease management. The British Society of Gastroenterology consensus guidelines on the management of IBD recommend considering psychological therapies (such as cognitive behavioral therapy, hypnotherapy, and mindfulness meditation) as adjunctive treatments to enhance symptom control and overall quality of life [42]. A practical implication of these findings is the suggestion for clinicians to explore personality traits and general psychopathology in IBD patients, considering UC and CD as different profiles and adopting a multidisciplinary approach in gastroenterology, offering a valuable opportunity to enhance the therapeutic strategy not only in terms of diagnosis, but also in prognostic aspects, in order to facilitate tailored intervention programs. Building on the present findings, future studies should consider recruiting UC and CD patients through systematic procedures and on a larger scale. This would allow for more robust statistical power and provide psychiatrists and clinical psychologists with new insights into the management of these chronic conditions. Moreover, it would be important to include a control group of healthy individuals, as well as a comparison group of patients affected by another chronic condition not related to IBD, in order to determine whether the observed psychological characteristics are specific to IBD or reflect more general responses to chronic illness.

## Figures and Tables

**Figure 1 biomedicines-13-01694-f001:**
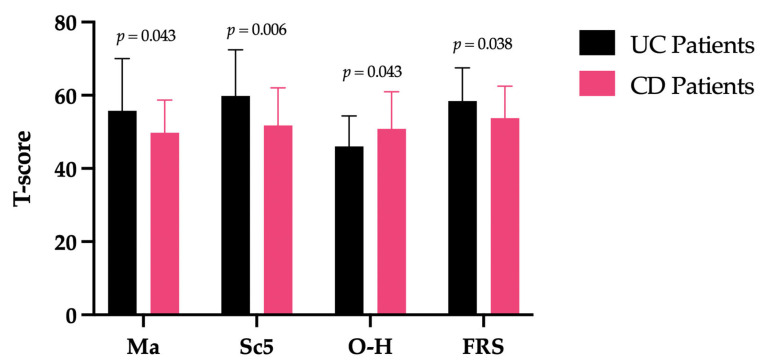
Differences in the mean scores of the MMPI-2 scales where a statistically significant difference was found between UC and CD patients. Note: CD (Crohn’s Disease); FRS (fears); Ma (hypomania); O-H (overcontrolled hostility); Sc5 (Lack of ego mastery, defective inhibition); UC (ulcerative colitis).

**Table 1 biomedicines-13-01694-t001:** Sociodemographic and clinical data differences between UC and CD patients.

	UC Patients	CD Patients
**Number**	29 (44.61%)	36 (55.39%)
**Male/Female ratio**	21/8 (72.41–27.58%)	17/19 (47.22–52.77%)
**Mean age ± SD**	42.8 ± 15.2	43.7 ± 14.8
**Disease activity**	Active = 14 (48.27%) Inactive = 12 (41.37%) In remission = 3 (10.34%)	Active = 23 (63.88%) Inactive = 8 (22.22%) In remission = 5 (13.88%)
**Years of illness**	Early-onset = 13 (44.82%) Late-onset = 16 (55.17%)	Early-onset = 16 (44.44%) Late-onset = 20 (55.55%)
**Biological therapy**	Yes = 14 (48.27%) No = 15 (51.72%)	Yes = 17 (47.22%) No = 19 (52.77%)

Note: CD (Crohn’s Disease); SD (Standard Deviation); UC (Ulcerative Colitis).

**Table 2 biomedicines-13-01694-t002:** Differences in the mean score of the MMPI-2 scales between UC and CD patients.

	UC Patients (Mean Score + SD)	CD Patients (Mean Score + SD)	F_(df)_	*p* Value
**Ma**	55.76 + 14.25	49.78 + 8.89	4.281_(1)_	0.043
**Sc5**	59.82 + 12.60	51.78 + 10.26	8.028_(1)_	0.006
**O-H**	46.00 + 8.35	50.83 + 10.11	4.271_(1)_	0.043
**FRS**	58.45 + 9.05	53.75 + 8.78	4.473_(1)_	0.038

Note: CD (Crohn’s Disease); df (degrees of freedom); FRS (fears); Ma (hypomania); O-H (overcontrolled hostility); Sc5 (lack of ego mastery, defective inhibition); SD (standard deviation); UC (ulcerative colitis).

## Data Availability

The original contributions presented in this study are included in the article/Appendix A. Further inquiries can be directed to the corresponding author.

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
