# Peer review of "Psychological Profiles in Ulcerative Colitis and Crohn’s Disease: Distinct Emotional and Behavioral Patterns"

_biomedicines, 2025, doi:10.3390/biomedicines13071694_

Round 1
Reviewer 1 Report
Comments and Suggestions for Authors
The manuscript entitled “Psychological Profiles in Ulcerative Colitis and Crohn's 2 Disease: Distinct Emotional and Behavioral Patterns” is an interesting topic. However, there are still several parts that need to be revised:
1. The sample size in the study is relatively small, which may affect the statistical efficacy and universality of the results. It is suggested that the author clearly explain this limitation in the discussion part, and suggest that the sample size should be expanded in future research to verify the current findings.
2. Cross-sectional design was adopted in the study, and the psychological characteristics of patients were evaluated at only one time point. This design can't capture the changes of personality characteristics in the process of disease development. It is suggested that the author mention this limitation in the discussion and recommend the longitudinal design for future research.
3. There is a significant difference in the gender distribution between UC and CD patients (the proportion of males in UC group is higher, and the proportion of females in CD group is higher). Although the author conducted a covariate analysis, gender differences may still have an impact on the results. It is suggested that the potential influence of gender on psychological characteristics should be further discussed.
4. Although the study found significant differences in some scales between UC and CD patients, the clinical significance of some differences was not fully discussed. It is suggested that the author explain in more detail the practical significance of these differences to patients' quality of life or disease management.
5. The study did not include healthy controls or patients with other chronic diseases as controls, so it is impossible to determine whether the observed psychological characteristics are specific to patients with IBD. It is suggested that this limitation should be mentioned in the discussion, and that future studies should be included in the control group to enhance the explanatory power of the results.
Author Response
We sincerely thank the Reviewer for the time, attention, and thoughtful feedback provided during the evaluation of our manuscript. We greatly appreciate the constructive comments, which have helped us to improve the clarity, methodological transparency, and clinical relevance of our work. In response to each point raised, we have carefully revised the manuscript to address the suggested changes and clarifications. Below, we provide a detailed point-by-point response outlining the modifications made and the rationale behind them. We trust that these revisions have strengthened the manuscript and made it more suitable for publication.
[COMMENTS 1] The sample size in the study is relatively small, which may affect the statistical efficacy and universality of the results. It is suggested that the author clearly explain this limitation in the discussion part, and suggest that the sample size should be expanded in future research to verify the current findings.
[RESPONSE 1] Thank you for this helpful comment. We have clarified in the manuscript how the patients were recruited, specifying that no formal sample size calculation was performed and that participants were enrolled in a naturalistic manner, as detailed in the Patients and Study Design section. Additionally, we have addressed this point more explicitly in the Discussion, acknowledging the limited sample size as a potential limitation. To address this, we have included a paragraph on future perspectives at the end of the Discussion section, emphasizing the importance of adopting a systematic recruitment strategy in future studies. This approach would allow for the inclusion of a larger and more representative sample, thereby increasing statistical power and enhancing the generalizability of findings.
[COMMENTS 2] Cross-sectional design was adopted in the study, and the psychological characteristics of patients were evaluated at only one time point. This design can't capture the changes of personality characteristics in the process of disease development. It is suggested that the author mention this limitation in the discussion and recommend the longitudinal design for future research.
[RESPONSE 2] Thank you for this insightful comment. As addressed in other responses, we have clarified in the Discussion that the MMPI-2 is designed to assess enduring trait-level personality characteristics, which tend to remain stable over time, even in the presence of changes in disease activity. Therefore, administering the MMPI-2 at different short-term intervals (e.g., 1 month, 3 months, 6 months, or 1 year) would likely yield comparable results. That said, we have reformulated the limitation related to the single time point to better reflect this methodological nuance. Specifically, we now emphasize that a second or third assessment point would be most meaningful after longer time frames (i.e., several years)—particularly in situations where the disease has stabilized or the patient has adapted to significant clinical changes (e.g., following surgery or stoma placement).
[COMMENTS 3] There is a significant difference in the gender distribution between UC and CD patients (the proportion of males in UC group is higher, and the proportion of females in CD group is higher). Although the author conducted a covariate analysis, gender differences may still have an impact on the results. It is suggested that the potential influence of gender on psychological characteristics should be further discussed.
[RESPONSE 3] We have conducted a MANCOVA (covariates: age and gender) on MMPI-2’s scales in order to control the possible influence of age and gender on our results. Moreover, the MMPI‑2 was scored using gender‑specific normative data, further minimizing any residual gender bias in scale interpretation. However, in the Limitations Section, we have added a paragraph acknowledging that, despite the non‑significant interaction, we suggest that further research could be designed with more gender‑balanced groups to confirm our results. This concept is clarified in these lines: “Even if we performed a MANCOVA to control the effect of gender on the MMPI‑2 scales, nevertheless we acknowledge that future studies would benefit from more gender‑balanced cohorts of UC/CD to fully validate these findings”.
[COMMENTS 4] Although the study found significant differences in some scales between UC and CD patients, the clinical significance of some differences was not fully discussed. It is suggested that the author explain in more detail the practical significance of these differences to patients' quality of life or disease management.
[RESPONSE 4] We thank the Reviewer for this important observation. In response, we have carefully revisited and expanded the Discussion section to better contextualize the clinical significance of the observed differences between UC and CD patients. Specifically, we now highlight how distinct psychological profiles—such as increased anxiety and phobic responses in UC and emotional inhibition in CD—may differentially influence patients’ quality of life, patterns of healthcare utilization, and disease self-management. We discuss how anxiety-related traits may lead to both delayed and unscheduled symptom-driven consultations, whereas suppressed emotional expression may hinder appropriate help-seeking, potentially compromising disease monitoring and treatment adherence. These insights have been integrated to underscore the relevance of psychological assessment in optimizing individualized care strategies for IBD patients.
[COMMENTS 5] The study did not include healthy controls or patients with other chronic diseases as controls, so it is impossible to determine whether the observed psychological characteristics are specific to patients with IBD. It is suggested that this limitation should be mentioned in the discussion, and that future studies should be included in the control group to enhance the explanatory power of the results.
[RESPONSE 5] Thank you for this valuable comment. We have revised the Discussion section to explicitly acknowledge the absence of a control group—such as healthy individuals or patients with other chronic conditions—as a limitation of the study. This omission prevents us from determining whether the psychological characteristics observed are specific to patients with IBD. In addition, we have expanded the section on future perspectives, recommending that future studies include appropriate control groups to enhance the explanatory power and generalizability of the findings.
Reviewer 2 Report
Comments and Suggestions for Authors
- Table 1 and Table 2, please revise the format of the table following the instructions of manuscript preparation of the journal;
- Figure 1, name for Y-axis is missing, use abbreviations for figure legends, i.e., UC patients, CD patients. Provide P value for each MMPI-2 scale mentioned in the figure;
- The inclusion of data related to blood levels of inflammatory cytokines, stress-related hormones as well as serotonin and their correlations with the MMPI-2 scales that significantly differ between UC and CD patients (e.g., scales in Figure 1 and Table 2).
Author Response
We would like to express our sincere gratitude to the Reviewer for the careful reading of our manuscript and for the constructive and insightful comments. Your suggestions have contributed significantly to improving the overall quality, clarity, and scientific rigor of the paper. We have addressed each point raised with great attention and have revised the manuscript accordingly. Below we provide a detailed point-by-point response, highlighting the modifications made to the tables, figure, and manuscript text, as well as clarifications regarding data availability and study limitations. We are confident that the revised version more clearly conveys the methodological choices and the clinical relevance of our findings.
[COMMENTS 1] Table 1 and Table 2, please revise the format of the table following the instructions of manuscript preparation of the journal;
[RESPONSE 1] Thank you for your suggestion. We have revised Table 1 and Table 2 according to the formatting guidelines provided in the journal’s instructions for manuscript preparation.
[COMMENTS 2] Figure 1, name for Y-axis is missing, use abbreviations for figure legends, i.e., UC patients, CD patients. Provide P value for each MMPI-2 scale mentioned in the figure;
[RESPONSE 2] Thank you for your valuable feedback. We have updated Figure 1 as recommended: the Y-axis label has been added, abbreviations such as “UC patients” and “CD patients” are now used in the figure legend, and p-values have been provided for each MMPI-2 scale included in the figure.
[COMMENTS 3] The inclusion of data related to blood levels of inflammatory cytokines, stress-related hormones as well as serotonin and their correlations with the MMPI-2 scales that significantly differ between UC and CD patients (e.g., scales in Figure 1 and Table 2).
[RESPONSE 3] It is not entirely clear whether the reviewer is suggesting the inclusion of such data or simply highlighting its potential relevance. However, if the suggestion is to add correlations between MMPI-2 scales and biological markers such as inflammatory cytokines, stress-related hormones, or serotonin levels, we regret to inform that these data are not available in our dataset. Our study includes only clinical variables, such as disease activity and current therapy (biologic vs. non-biologic). We have updated the limitations section of the manuscript to reflect this point. Specifically, we have acknowledged the absence of laboratory data—such as inflammatory cytokines, stress-related hormones—as a limitation, given its relevance to the interpretation of psychopathological features.
Reviewer 3 Report
Comments and Suggestions for Authors
Dear Author
Please consider the following suggestions and recommendations for the manuscript titled "Psychological Profiles in Ulcerative Colitis and Crohn's Disease: Distinct Emotional and Behavioral Patterns."
Section 1. Introduction
Lines 49 to 51: Revise the sentence describing the prevalence estimates. It is incomplete in the text.
Line 53: "The pooled prevalence..." - state which sources or what diseases were reviewed as contributors to the collective "pool" of statistics. The reader should not be assuming that IBD sources were considered. Add a reference for this statement.
Line 61: MMPI-2 has not been abbreviated in previous text.
Line 62 - 1: Reference West et al., 1970 is incorrect. This reference has one author.
Line 62 - 2: Remove duplicate reference - West, 1970.
Lines 62 to 63: Ulcerative colitis has been abbreviated in previous text - see line 45.
Line 77: Revise order of references.
Lines 82 to 92: This text forms part of the literature review. Consider placing text describing the aims of the study i.e. lines 80 to 82 - aim one and lines 92 to 95 - aim two, after the completed literature review and introduction.
Section 2. Materials and Methods
For context of the state of patients, state the dates of interviews and assessments. Considering the psychological effects of the COVID pandemic, time frames are important for understanding and may add insight to emotion and personality investigations.
Lines 125 to 126: Did the authors of this study validate the MMPI-2 method?
If so, the validation data must be provided.
If not, this statement must be referenced.
Section 3. Results
Table 1 can be revised into a smaller version.
Author Response
We sincerely thank the Reviewer for the thorough and constructive evaluation of our manuscript. Your detailed comments and careful attention to methodological, editorial, and structural aspects have been invaluable in improving the clarity, rigor, and overall quality of the work. We have carefully addressed each point raised and implemented the necessary revisions throughout the manuscript, including clarifications in the Introduction, corrections to references and abbreviations, updates to the Methods section, and reorganization of Tables and Figures in accordance with journal guidelines. Below we provide a point-by-point response to each comment, detailing the corresponding changes made to the manuscript.
Section 1. Introduction
[COMMENTS 1] Lines 49 to 51: Revise the sentence describing the prevalence estimates. It is incomplete in the text.
[RESPONSE 1] Thank you for your comment. We have revised the sentence to ensure clarity and completeness
[COMMENTS 2] Line 53: "The pooled prevalence..." - state which sources or what diseases were reviewed as contributors to the collective "pool" of statistics. The reader should not be assuming that IBD sources were considered. Add a reference for this statement.
[RESPONSE 2] Thank you for your comment. As addressed in the previous response, we have completely rewritten that section to ensure greater clarity and precision. The revised text explicitly states that the data refer to patients with inflammatory bowel disease (IBD) and attributes the findings to the systematic review by Neuendorf et al.: “According to a systematic review by Neuendorf et al., the pooled prevalence of anxiety disorders in patients with inflammatory bowel disease (IBD) was 20.5%, while the prevalence of anxiety symptoms reached 35.1%. The prevalence was significantly higher in patients with active disease (75.6%) compared to those in remission. Similarly, the pooled prevalence of depressive disorders was 15.2%, and 21.6% for depressive symptoms, with higher rates observed in Crohn’s disease (25.3%) compared to ulcerative colitis, and in patients with active disease (40.7%) versus those in remission.” The source has been clearly referenced, and it is now evident that the pooled data are specific to IBD populations.
[COMMENTS 3] Line 61: MMPI-2 has not been abbreviated in previous text.
[RESPONSE 3] Thank you for your observation. We have corrected the text by introducing the full name of the MMPI-2 at first mention and including the abbreviation accordingly.
[COMMENTS 4] Line 62 - 1: Reference West et al., 1970 is incorrect. This reference has one author.
[RESPONSE 4] Thank you for pointing that out. The reference has been corrected to accurately reflect the single author attribution.
[COMMENTS 5] Line 62 - 2: Remove duplicate reference - West, 1970.
[RESPONSE 5] Thank you for your comment. The duplicate reference appears due to a specific reference style format, which we have now adjusted in accordance with the journal’s guidelines.
[COMMENTS 6] Lines 62 to 63: Ulcerative colitis has been abbreviated in previous text - see line 45.
[RESPONSE 6] Thank you for your observation. We have corrected the text to ensure consistent use of the abbreviation “UC” for ulcerative colitis, as introduced earlier in the manuscript.
[COMMENTS 7] Line 77: Revise order of references.
[RESPONSE 7] Thank you for the suggestion. The reference order will be revised throughout the manuscript to comply with the journal’s formatting guidelines.
[COMMENTS 8] Lines 82 to 92: This text forms part of the literature review. Consider placing text describing the aims of the study i.e. lines 80 to 82 - aim one and lines 92 to 95 - aim two, after the completed literature review and introduction.
[RESPONSE 8] Thank you for the suggestion. We have reorganized the text accordingly and placed the study objectives at the end of the Introduction section, following the completion of the literature review.
Section 2. Materials and Methods
[COMMENTS 9] For context of the state of patients, state the dates of interviews and assessments. Considering the psychological effects of the COVID pandemic, time frames are important for understanding and may add insight to emotion and personality investigations.
[RESPONSE 9] Thank you for the observation. We have updated the Materials and Methods section by specifying that patient recruitment took place between 2018 and 2019.
[COMMENTS 10] Lines 125 to 126: Did the authors of this study validate the MMPI-2 method? If so, the validation data must be provided. If not, this statement must be referenced.
[RESPONSE 10] We did not perform an independent validation of the MMPI‑2 within our sample. As a consequence, we have added the following reference in the Methods section to clarify this point: “We did not perform an independent validation of the MMPI‑2 within our sample and the scales were administered and scored in accordance with the procedures and normative data described by Butcher et al. (2001).”
Section 3. Results
[COMMENTS 11] Table 1 can be revised into a smaller version.
[RESPONSE 11] We have revised Table 1 in accordance with the journal’s guidelines and formatting template, which also allowed us to present it in a more concise and compact format.
Reviewer 4 Report
Comments and Suggestions for Authors
Journal
Biomedicines (ISSN 2227-9059)
Manuscript ID
biomedicines-3666857
Type: Article
Title
Psychological Profiles in Ulcerative Colitis and Crohn's Disease: Distinct Emotional and Behavioral Patterns
Section
Molecular and Translational Medicine
Special Issue
Crohn's Disease and Ulcerative Colitis: From Pathophysiology to Novel Therapeutic Approaches (3rd Edition)
___________________________________
OVERALL COMMENTS
In this manuscript, the authors intended to delineate the personality characteristics of a sample of patients affected by inflammatory bowel disease and to investigate the differences between individuals with ulcerative colitis (UC) and Crohn's disease (CD). For that purpose, they included enrolled 29 UC patients and 36 CD patients. Each participant completed the Minnesota Multiphasic Personality Inventory-2 (MMPI-2), which was subsequently scored and interpreted by trained psychologists. Their results showed that the total sample showed clinically significant elevations in Hypochondriasis (Hs), Health Concerns (HEA), General Health Concerns (HEA3), and Physical Malfunctioning (D3) scales. UC patients had statistically significantly higher scores on Hypomania (p=0.043), Lack of Ego Mastery - Defective Inhibition (p=0.006), and Fears (p=0.038) scales than CD patients. On the other hand, CD patients showed statistically significant higher scores on the Overcontrolled Hostility scale (p=0.043). The conclusions showed that UC and CD patients have emotional difficulties linked to clinical conditions. These aspects appear to be accompanied by shifts in mood towards a more depressive state.
Important comment: Please see MDPI guidelines for citations of the references. I suggest that the authors check it along with all the text and observe the references section.
TITLE
The title is adequate.
ABSTRACT
The authors need to check punctuation in some places in the abstract and the text, such as missing commas. Also, some other small errors.
The authors start the abstract by calling the section “Background”. However, the sentence “We wanted to delineate the personality characteristics of a sample 21 of patients affected by inflammatory bowel disease and to investigate the differences between individuals with ulcerative colitis (UC) and Crohn's disease (CD)” is not a background. It is much more the Objective.
Please allow me to suggest that the abstract should be like this:
Background: Please include a background related to the title of the study. Objective/aims: This study aimed to delineate the personality characteristics of a sample of patients affected by inflammatory bowel disease and investigate the differences between individuals with ulcerative colitis (UC) and Crohn's disease (CD). Methods: We enrolled 29 (44.61%) UC patients and 36 (55.39%) CD patients, all aged at least 18 years. Each participant completed the Minnesota Multiphasic Personality Inventory-2 (MMPI-2), which was subsequently scored and interpreted by trained psychologists. The MMPI-2 is a 567-item inventory with dichotomous answers (true or false), providing measures of a wide range of symptoms, beliefs, attitudes, and personality characteristics. Results: The total sample showed clinically significant elevations in Hypochondriasis (Hs), Health Concerns (HEA), General Health Concerns (HEA3), and Physical Malfunctioning (D3) scales. UC patients had significantly higher scores on Hypomania (p=0.043), Lack of Ego Mastery - Defective Inhibition (p=0.006), and Fears (p=0.038) scales than CD patients. On the other hand, CD patients showed statistically significant higher scores on the Overcontrolled Hostility scale (p=0.043). Conclusions: Both these groups of patients experienced emotional difficulties linked to clinical conditions, resulting in an increased 35 focus on the body and illnesses. These aspects appear to be accompanied by shifts in mood towards a more depressive state. Notably, the UC group demonstrates a greater degree of impairment compared to the CD group, with experiences of anxiety, stress, difficulties in emotional control, and emerging relational challenges.
_______
KEYWORDS
The authors presented the following keywords:
Personality; Ulcerative Colitis; Crohn’s Disease; Inflammatory Bowel Disease; 40 Psychopathology; Psychosomatic.
I suggest:
Ucerative Colitis; Crohn’s Disease; Inflammatory Bowel Disease; Emotional Patterns; Behavioral Pattern
INTRODUCTION
Please see MDPI guidelines for citations of the references. Furthermore, can the authors find a reference published in 2024 or 2025 to follow this section? The cited references are too old. There are a plethora of incredible good papers on PUBMED.com. Please make a search and update your references.
_______
METHODS
In lines 100-102 we can read that “We recruited consecutive patients affected by UC and CD who were at least 18 years old. Inclusion criteria were age > 18, diagnosis of UC/CD, at least a level of education of lower secondary school diploma, and acceptance of the informed consent.”
I suggest that the authors mention where the patients were recruited from.
Please include a flowchart for the invited participants, including the exclusion/inclusion criteria, how many participants were selected, and how many participants remained until the end of the study.
Why did the authors use the MMPI-2 (old version) when the MMPI-3 was already available?
Different criteria for disease activity were used: Different indices were used (Mayo for UC, Harvey-Bradshaw for CD). Wouldn't this be a limitation for this study?
The criteria included in the study were investigated at a single time point, making it impossible to make causal inferences such as active disease. Wouldn't this be an important limitation to be considered?
The sample used in this study is not large. Was there a sample size calculation for this study? In lines 338-340 we can read that “This study presents some limitations. Sample size of UC and CD patients is relatively small; in future research, it would be interesting to analyze personality on a larger cohort of patients with IBD.” Please include here in the Methods section that there was or was not a sample size calculation.
Ethical concerns: none. The authors included the ethical approval.
RESULTS
Include the definitions for all the abbreviations used in the Figure and table. The word “Figure” does not need to be in italics.
_________
DISCUSSION
This section is long. However, the concern is: why did the authors not use newer references? Please visit PUBMED.com.
Furthermore, this section seems to go beyond the obtained results, suggesting causal relationships or mechanisms (e.g., “mood shifts toward a more depressed state”). Does the cross-sectional design support this interpretation?
In lines 320-322 we can read that “This result is in contrast with a previous study in which it was highlighted that CD patients presented higher impulsive sensation seeking than UC patients (Hyphantis et al., 2010). “ Please comment this contrast.
CONCLUSION
Included at the end of the Discussion section.
________
I suggest including a section with the FUTURE PERSPECTIVES for this study:
Include the future perspectives for this study.
How can this review contribute to further research?
Include recommendations on how to standardize this model for future applications.
____________
REFERENCES
As mentioned before, please include references published in the last few years.
Author Response
We sincerely thank the Reviewer for the thoughtful and detailed feedback provided on our manuscript. Your comments have been extremely helpful in refining the quality, clarity, and scientific rigor of our work. We deeply appreciate the time you dedicated to reviewing our study and offering valuable suggestions regarding content, structure, citations, and methodological transparency. In response, we have carefully revised the manuscript in accordance with your recommendations, including updates to the Abstract, Introduction, Methods, Results, Discussion, and Reference sections, as well as improved compliance with the MDPI formatting guidelines. Below, we provide a point-by-point reply to each of your comments, outlining the corresponding changes made to the manuscript.
[COMMENTS 1] Important comment: Please see MDPI guidelines for citations of the references. I suggest that the authors check it along with all the text and observe the references section.
[RESPONSE 1] We thank the Reviewer for this helpful reminder. In accordance with the MDPI guidelines, we have thoroughly reviewed the entire manuscript and revised all in-text citations and the reference list to ensure full compliance with the journal’s formatting requirements.
[COMMENTS 2] The authors need to check punctuation in some places in the abstract and the text, such as missing commas. Also, some other small errors.
[RESPONSE 2] We have carefully reviewed the abstract and main text, and we have corrected punctuation errors, including missing commas, as well as other minor issues throughout the manuscript.
[COMMENTS 3] The authors start the abstract by calling the section “Background”. However, the sentence “We wanted to delineate the personality characteristics of a sample 21 of patients affected by inflammatory bowel disease and to investigate the differences between individuals with ulcerative colitis (UC) and Crohn's disease (CD)” is not a background. It is much more the Objective.
[RESPONSE 3] Thank you for your observation, and we apologize for the oversight. The section has been completely revised and appropriately renamed “Background/Objectives” to accurately reflect its content. The revision was made in accordance with the journal’s guidelines and the formatting provided in the official template.
[COMMENTS 4] Background: Please include a background related to the title of the study.
Objective/aims: This study aimed to delineate the personality characteristics of a sample of patients affected by inflammatory bowel disease and investigate the differences between individuals with ulcerative colitis (UC) and Crohn's disease (CD). Methods: We enrolled 29 (44.61%) UC patients and 36 (55.39%) CD patients, all aged at least 18 years. Each participant completed the Minnesota Multiphasic Personality Inventory-2 (MMPI-2), which was subsequently scored and interpreted by trained psychologists. The MMPI-2 is a 567-item inventory with dichotomous answers (true or false), providing measures of a wide range of symptoms, beliefs, attitudes, and personality characteristics. Results: The total sample showed clinically significant elevations in Hypochondriasis (Hs), Health Concerns (HEA), General Health Concerns (HEA3), and Physical Malfunctioning (D3) scales. UC patients had significantly higher scores on Hypomania (p=0.043), Lack of Ego Mastery - Defective Inhibition (p=0.006), and Fears (p=0.038) scales than CD patients. On the other hand, CD patients showed statistically significant higher scores on the Overcontrolled Hostility scale (p=0.043). Conclusions: Both these groups of patients experienced emotional difficulties linked to clinical conditions, resulting in an increased 35 focus on the body and illnesses. These aspects appear to be accompanied by shifts in mood towards a more depressive state. Notably, the UC group demonstrates a greater degree of impairment compared to the CD group, with experiences of anxiety, stress, difficulties in emotional control, and emerging relational challenges.
[RESPONSE 4] Thank you for your comment. As noted in our previous response, we have revised the section accordingly. A proper background related to the title of the study has been added, and the section has been renamed “Background/Objectives”in compliance with the journal's guidelines and formatting template. The updated section now clearly distinguishes between the context of the study and its aims.
[COMMENTS 5 ] KEYWORDS: The authors presented the following keywords: Personality; Ulcerative Colitis; Crohn’s Disease; Inflammatory Bowel Disease; 40 Psychopathology; Psychosomatic. I suggest: Ucerative Colitis; Crohn’s Disease; Inflammatory Bowel Disease; Emotional Patterns; Behavioral Pattern
[RESPONSE 5] Thank you for the suggestion. We have updated the keywords as recommended, replacing the original terms with: Ulcerative Colitis; Crohn’s Disease; Inflammatory Bowel Disease; Emotional Patterns; Behavioral Pattern.
[COMMENTS 6] INTRODUCTION: Please see MDPI guidelines for citations of the references.
[RESPONSE 6] We thank the Reviewer for this helpful reminder. In accordance with the MDPI guidelines, we have thoroughly reviewed the entire manuscript and revised all in-text citations and the reference list to ensure full compliance with the journal’s formatting requirements.
[COMMENTS 7] Furthermore, can the authors find a reference published in 2024 or 2025 to follow this section? The cited references are too old. There are a plethora of incredible good papers on PUBMED.com. Please make a search and update your references.
[RESPONSE 7] Thank you for your suggestion to include more recent references. We agree that there are good papers available on PubMed; unfortunately, in addition to those we have already cited, there are no other published studies specifically addressing personality in our target populations. Existing research primarily focuses on depressive or anxiety symptoms, while personality assessment in patients with UC and CD remains a largely unexplored area in the current literature. This important gap actually represents the principal strength of our study. There is no available literature in 2024-2025 specifically addressing this topic. However, we identified and added a relevant study (Jordi et al., 2022)
[COMMENTS 8] METHODS: In lines 100-102 we can read that “We recruited consecutive patients affected by UC and CD who were at least 18 years old. Inclusion criteria were age > 18, diagnosis of UC/CD, at least a level of education of lower secondary school diploma, and acceptance of the informed consent.” I suggest that the authors mention where the patients were recruited from.
[RESPONSE 8] Thank you for the suggestion. We have updated the Methods section to specify the recruitment site, indicating that patients were enrolled at the IBD Unit of CEMAD, the Center for Digestive Disease at the A. Gemelli IRCCS Hospital in Rome.
[COMMENTS 9] Please include a flowchart for the invited participants, including the exclusion/inclusion criteria, how many participants were selected, and how many participants remained until the end of the study.
[RESPONSE 9] Thank you for this comment, which allows us to provide some important clarifications. Patients were enrolled in a naturalistic manner: they were referred to the psychologist of the Psychogastroenterology Service either upon their own request (as part of the visit package routinely offered by our center) or when considered appropriate by the gastroenterologist. As a result, we did not systematically record the number of patients who declined to complete the MMPI-2, and therefore it is not possible to construct a flowchart showing the invited and excluded participants. We had already specified at the end of the “Patients and Study Design” paragraph that the study population was defined based on available data from patients meeting the inclusion criteria, reflecting real-world conditions rather than a predetermined sample size. Nonetheless, this is an appropriate and constructive comment that prompted us to revise the beginning of this section to improve the methodological clarity and enhance the reproducibility of the study for readers and future researchers.
[COMMENTS 10] Why did the authors use the MMPI-2 (old version) when the MMPI-3 was already available?
[RESPONSE 10] We enrolled patients between 2018 and 2019 and in these years MMPI-3 was not already available, because it was published in 2020.
[COMMENTS 11] Different criteria for disease activity were used: Different indices were used (Mayo for UC, Harvey-Bradshaw for CD). Wouldn't this be a limitation for this study?
[RESPONSE 11] Using different disease activity scores—Mayo for UC and Harvey–Bradshaw for CD—is not a limitation but actually a strength of the study. Each index is the gold standard for its specific condition, offering accurate and clinically valid assessment of disease activity. Since our goal was to explore psychological and personality differences—not to directly compare disease activity across UC and CD—using the most appropriate index for each group ensures accurate clinical characterization and enhances validity. Moreover, disease activity was included solely for descriptive purposes. Correlating it with specific MMPI‑2 scales falls outside the scope of this study, especially considering that the MMPI‑2 is designed to assess stable personality traits and enduring psychopathological characteristics, not transient states like disease flares or remission phases
[COMMENTS 12] The criteria included in the study were investigated at a single time point, making it impossible to make causal inferences such as active disease. Wouldn't this be an important limitation to be considered?[RESPONSE 12] Thank you for this comment, which gives us the opportunity to offer some clarifications. As mentioned in other responses, the MMPI‑2 is designed to capture enduring trait-level characteristics, meaning that administering the test at different time points—such as after 1 month, 3 months, 6 months, or even 1 year—would likely yield substantially comparable results, even in the presence of changes in disease activity. That said, we agree that the single time point of assessment is still a methodological aspect worth addressing. For this reason, we have expanded and refined the discussion of this limitation in the appropriate section of the manuscript.
[COMMENTS 13] The sample used in this study is not large. Was there a sample size calculation for this study? In lines 338-340 we can read that “This study presents some limitations. Sample size of UC and CD patients is relatively small; in future research, it would be interesting to analyze personality on a larger cohort of patients with IBD.” Please include here in the Methods section that there was or was not a sample size calculation.
[RESPONSE 13] Thank you for this insightful comment. As noted at the end of the Patients and Study Design section, the study population was defined based on available data from patients meeting the inclusion criteria, reflecting real-world conditions rather than a predetermined sample size. Therefore, no formal sample size calculation was performed. We have now clarified this explicitly in the Methods section. Furthermore, based on the experience gained from this work, we have added the following note at the end of the paper: "Building on the present findings, future studies should consider recruiting UC and CD patients through systematic procedures and on a larger scale. This would allow for more robust statistical power and provide psychiatrists and clinical psychologists with new insights into the management of these chronic conditions."
[COMMENTS 14] RESULTS: Include the definitions for all the abbreviations used in the Figure and table. The word “Figure” does not need to be in italics.
[RESPONSE 14] Thank you for your comment. All figure and table captions have been carefully reviewed, and the abbreviations have been fully defined and listed in alphabetical order to facilitate consultation. Additionally, the word "Figure" is no longer italicized, in accordance with the journal’s formatting guidelines.
[COMMENTS 15] DISCUSSION: This section is long. However, the concern is: why did the authors not use newer references? Please visit PUBMED.com.
[RESPONSE 15] We agree with the reviewer’s comment, unfortunately there are no other published studies specifically addressing personality in patients with UC, CD or IBD in general. We found it quite surprising that research on personality in individuals with IBD is so quite limited and dated - not only in terms of the use of the MMPI, which is a validated and widely recognized assessment instrument, but also in terms of broader investigations on personality traits in this clinical population. Personality assessment in patients with UC and CD remains a largely unexplored area in the current literature, as a consequence this important gap actually represents the principal strength of our study.
[COMMENTS 16] Furthermore, this section seems to go beyond the obtained results, suggesting causal relationships or mechanisms (e.g., “mood shifts toward a more depressed state”). Does the cross-sectional design support this interpretation?
[RESPONSE 16] We applied the standard interpretive norms of the MMPI-2 to describe the personality profiles observed in our sample, and we agree that this approach does not establish causality. Our use of terms such as “mood shifts toward a more depressed state” was intended to reflect an association identified by the MMPI-2 patterns, not a demonstrated causal mechanism. However, we clarified this aspect in the manuscript.
[COMMENTS 17] In lines 320-322 we can read that “This result is in contrast with a previous study in which it was highlighted that CD patients presented higher impulsive sensation seeking than UC patients (Hyphantis et al., 2010). “ Please comment this contrast.
[RESPONSE 17] Thank you for this important observation. We agree that the contrast between our findings and those reported by Hyphantis et al. (2010) warrants further consideration. This contrast has now been appropriately addressed in the manuscript, immediately following the sentence cited by the reviewer. Specifically, we discuss how the two studies assess different psychological constructs—emotional vulnerability and disorganization (Sc5) versus behavioral impulsivity and sensation seeking—and how methodological differences may account for the divergent findings.
[COMMENTS 18] CONCLUSION: Included at the end of the Discussion section. I suggest including a section with the FUTURE PERSPECTIVES for this study: Include the future perspectives for this study. How can this review contribute to further research? Include recommendations on how to standardize this model for future applications.
[RESPONSE 18] Thank you for your valuable suggestion. As noted in response to another comment, we have added a brief section addressing future perspectives at the end of the Discussion. This addition highlights how the experience gained from the present study—based on naturalistic patient recruitment—may guide the design of future research using systematic recruitment methods. Such an approach would allow for the inclusion of a larger and more representative sample, helping to standardize the model and produce more generalizable findings. We believe this will contribute meaningfully to future research on the psychological and personality profiles of patients with IBD.
[COMMENTS 19] REFERENCES: As mentioned before, please include references published in the last few year
[RESPONSE 19] As we explained above, unfortunately there is no recent literature specifically addressing this topic. However, we added a relevant study (Jordi et al., 2022) about IBD and depression and another one that explore the impact of IBD on quality of life Gabova et al. (2024).
Round 2
Reviewer 2 Report
Comments and Suggestions for Authors
The revised manuscript improved in data presentation and points of discussion and can be considered for publication by the journal. I have no further comments.
Reviewer 4 Report
Comments and Suggestions for Authors
Dear authors,
Thank you for responding to my comments and suggestions.
With my best regards.